# Predictive and Prognostic Value of Serum Neutrophil Gelatinase-Associated Lipocalin for Contrast-Induced Acute Kidney Injury and Long-Term Clinical Outcomes after Percutaneous Coronary Intervention

**DOI:** 10.3390/jcm11195971

**Published:** 2022-10-10

**Authors:** Jaeho Byeon, Ik Jun Choi, Dongjae Lee, Youngchul Ahn, Mi-Jeong Kim, Doo Soo Jeon

**Affiliations:** Division of Cardiology, Department of Internal Medicine, Incheon St. Mary’s Hospital, College of Medicine, The Catholic University of Korea, Seoul 06591, Korea

**Keywords:** NGAL, contrast-induced acute kidney injury, coronary artery disease, percutaneous coronary intervention

## Abstract

Neutrophil gelatinase-associated lipocalin (NGAL) has been proposed as an early marker for estimating the risk of contrast-induced acute kidney injury (CI-AKI). However, the predictive value of baseline serum NGAL levels for CI-AKI remains unclear. Serum NGAL was measured before percutaneous coronary intervention in 633 patients with coronary artery disease. The primary clinical endpoints were a composite of major adverse cardiac and cerebrovascular events (MACCEs; cardiac death, myocardial infarction, stroke, and any revascularization). The mean follow-up duration was 29.4 months. Ninety-eight (15.5%) patients developed CI-AKI. Compared with patients without CI-AKI, baseline serum NGAL was higher in patients with CI-AKI (149.6 ± 88.8 ng/mL vs. 138.0 ± 98.6 ng/mL, *p* = 0.0279), although serum creatinine and estimated glomerular filtration rate were not different between groups. Patients in the highest tertile of baseline serum NGAL showed a significantly higher rate of MACCEs (10.5% vs. 3.8%, *p* = 0.02). Using the first tertile as a reference, the adjusted hazard ratios for MACCEs in patients in the second and third tertiles of NGAL were 2.151 (confidence interval (CI) 0.82 to 5.59, *p* = 0.116) and 2.725 (CI 1.05 to 7.05, *p* = 0.039), respectively. Baseline serum NGAL is a reliable marker for predicting CI-AKI, and high serum NGAL levels are associated with a higher incidence rate of long term MACCEs.

## 1. Introduction

Contrast-induced acute kidney injury (CI-AKI) is a major complication of coronary artery disease (CAD) treated by percutaneous coronary intervention (PCI) [1] and is associated with increased mortality and cardiovascular outcomes [2,3,4]. CI-AKI is characterized by a decline in kidney function that occurs within days after the intravascular administration of contrast medium [5]. The mechanisms involved in CI-AKI include ischemic injury to the renal medulla, oxidative damage, and direct toxicity involving the renal tubules. The prediction and prevention of CI-AKI are important in the management of the periprocedural period. Many studies have identified some biomarkers that may be used to anticipate the development of acute kidney injury in various clinical situations. Neutrophil gelatinase-associated lipocalin (NGAL) is a well-known marker of kidney tubular injury [6]. The predictive power of changes in NGAL for AKI after contrast use has been widely reported [7,8,9,10]. However, the predictive value of baseline serum NGAL for CI-AKI after PCI remains controversial. We sought to evaluate the ability of baseline serum NGAL to predict the incidence of CI-AKI and the prognostic performance in patients with CAD undergoing PCI.

## 2. Materials and Methods

### 2.1. Study Populations

We screened 796 consecutive patients with CAD scheduled for PCI at Incheon St. Mary’s Hospital between September 2015 and November 2017. Patients with cardiogenic shock, end-stage renal disease requiring dialysis, or insufficient blood samples and those who did not undergo PCI were excluded. Of the 796 patients, 633 had samples available for the measurement of the serum level of NGAL. All participants provided written informed consent to participate before PCI and blood sampling. The study protocol was reviewed and approved by the appropriate institutional review board.

### 2.2. PCI Procedure and Medical Treatments

Coronary angiography and PCI were performed according to standard techniques at the operator’s discretion. The contrast medium used was iodixanol (Visipaque, GE Healthcare, Chicago, IL, USA). Antiplatelet therapy and periprocedural anticoagulation were administered according to standard regimens. All patients were recommended for guideline-directed medical therapy, including antiplatelets, statins, beta-blockers, or renin-angiotensin-aldosterone blockades, following standard European and American guidelines [11,12]. Clinical follow-up was performed every 3 months after the index procedure.

### 2.3. Laboratory Measurements

Blood samples were drawn upon arrival at the catheterization laboratory and were collected immediately after sheath insertion and before PCI. After the blood was centrifuged, plasma was subsequently stored at −80 °C. Serum NGAL levels were measured by a Human Lipocalin-2/NGAL Quantikine ELISA kit (Catalog #DLCN20) from R&D Systems (Minneapolis, MN, USA). The measurement of NGAL levels was performed in the Clinical Research Laboratory, Incheon St. Mary’s Hospital, The Catholic University of Korea.

### 2.4. Study Endpoints and Definitions

The primary endpoint was major adverse cardiac and cerebrovascular events (MACCEs), including cardiovascular death, nonfatal myocardial infarction, nonfatal stroke, and any revascularization. Patient follow-up information, including survival and clinical events, was collected through hospital chart review and telephone interviews with patients by trained reviewers who were blinded to the study results. In addition, the mortality data were verified by the database of the National Health Insurance Corporation, Korea, using a unique personal identification number.

CI-AKI occurring within 72 h of contrast use is defined by the International Kidney Disease Improving Global Outcomes classification as follows: an increase in serum creatinine of ≥0.3 mg/dL, an increase in serum creatinine of ≥1.5 times baseline, a urine volume ≤0.5 mL/kg/h for 6 h. [13]

### 2.5. Statistical Analysis

Continuous variables are expressed as the mean ± standard deviation and were analyzed by independent sample *t* test or the Mann-Whitney *U* test. Categorical variables are presented as percentages or rates and were analyzed by the chi-square test or Fisher’s exact test. Serum NGAL levels are expressed as a continuous variable or by groups, categorized into three groups by tertiles. Differences in baseline characteristics between the different tertiles of serum NGAL levels were evaluated using one-way analysis of variance for continuous variables and the chi-square test for categorical variables. Traditional cardiovascular risk factors and CI-AKI risk factors were used for univariate analysis, and only variables with *p* < 0.1 in univariate analysis were analyzed with multivariate analysis for association with the risk of contrast-induced acute kidney injury. Multivariable analysis was performed to assess the prognostic value of serum NGAL and MACCEs after adjusting for age, sex, estimated glomerular filtration rate, body mass index, hypertension, diabetes mellitus, smoking, family history of coronary artery disease, chronic kidney disease, dyslipidemia, prior stroke, prior myocardial infarction, acute myocardial infarction, hypotension, multivessel disease, and left ventricular ejection fraction. Hazard ratios (HR) were estimated with multivariable adjusted Cox proportional hazards models, using the first tertile of NGAL as a reference. Kaplan-Meier curves were used to analyze the clinical outcomes and overall survival rate of patients. All analyses were 2-tailed, and *p* < 0.05 was considered indicative of statistical significance. All statistical analyses were performed using SPSS 27 statistical software (SPSS Inc., Chicago, IL, USA) and R version 4.2.1 (R Foundation for Statistical Computing, Vienna, Austria).

## 3. Results

### 3.1. Patient Characteristics

Overall, 633 patients with CAD treated by PCI were analyzed. The baseline characteristics of the total patient population per tertile are shown in Table 1. The mean age of all 633 patients was 65.5 ± 11.7 years old, and 66.0% of the patients were men. Among them, 225 patients (35.5%) had a history of chronic kidney disease, defined by an estimated glomerular filtration rate (eGFR) of less than 60 mL/min/1.73 m^2^. When categorized into three groups according to the tertile of baseline serum NGAL (NGAL tertiles, 1st: 25.4 to 83.7 ng/mL, 2nd: 83.8 to 143.8 ng/mL, 3rd: 143.9 to 567.9 ng/mL), there was a significant trend toward higher serum NGAL levels with older age, hypertension, diabetes, chronic kidney disease, low ejection fraction, high C-reactive protein, higher contrast volume, multivessel coronary disease, larger number of stents, and longer stent length. There were no significant differences with regard to sex, body mass index, dyslipidemia, or culprit coronary lesions among the three tertiles.

### 3.2. Serum NGAL Levels and Contrast-Induced Acute Kidney Injury

Among all patients, 98 (15.5%) patients developed CI-AKI (Appendix A). Those subjects who suffered from CI-AKI had higher baseline NGAL levels than those without CI-AKI (149.6 ± 88.8 ng/mL vs. 138.0 ± 98.6 ng/mL, *p* = 0.0279). However, the serum creatinine level (1.14 ± 1.53 mg/dL vs. 1.09 ± 0.63 mg/dL, *p* = 0.737) and eGFR (73.0 ± 34.2.4 mL/min/1.73 m^2^ vs. 71.5 ± 26.9 mL/min/1.73 m^2^, *p* = 0.685) were not different between the two groups. Additionally, there was no difference in the infused contrast volume (217.9 ± 121.0 mL vs. 220.8 ± 115.6 mL, *p* = 0.831). Patients who required renal replacement therapy were not reported during in-hospital periods or overall follow-up.

There was an increase in the incidence of CI-AKI with increasing tertiles of NGAL (Figure 1). Compared with the reference group (1st tertile), the adjusted odds ratios for CI-AKI were 2.7 (CI 1.391–5.239, *p* = 0.003) for the 2nd tertile of NGAL and 3.57 (CI 1.788–7.141, *p* < 0.001) for the 3rd tertile of NGAL (Table 2).

### 3.3. Serum NGAL Levels and Cardiac and Cerebrovascular Outcomes

The median follow-up duration was 29.4 months (IQR 23.8 to 37.2). During the overall follow-up, MACCEs occurred in 43 patients (6.8%). Cardiovascular death, nonfatal myocardial infarction, nonfatal stroke, and any revascularization occurred in 24 (3.79%), 2 (0.31%), 6 (0.94%), and 16 (2.52%) patients, respectively. The individual components of MACCEs and all-cause death are presented in Table 3, and Kaplan-Meier curves for serum NGAL levels according to tertiles and primary outcomes are shown in Figure 2. Patients in the highest tertile showed significantly higher rates of MACCEs (10.5% vs. 3.8%, *p* = 0.02) and all-cause death (11.0% vs. 1.4%, *p* < 0.001) than patients in the first tertile. Using the first tertile as a reference, the adjusted HRs for those in the second and third tertiles of NGAL were 2.151 (CI 0.827–5.592, *p* = 0.116) and 2.725 (CI 1.052–7.058, *p* = 0.039) for MACCEs and 3.692 (CI 0.938–14.522, *p* = 0.062) and 6.172 (CI 1.650–23.077, *p* = 0.007) for all-cause death (Table 4).

## 4. Discussion

The present study demonstrates that baseline serum NGAL levels can be used to predict the occurrence of CI-AKI independent of potential confounding factors such as serum creatinine, eGFR, and infused contrast volume. In addition, baseline serum NGAL levels are associated with MACCEs and all-cause mortality in patients with CAD treated with PCI. According to these findings, baseline serum NGAL might serve as a predictor of the development of CI-AKI and cardiac and cerebrovascular outcomes in patients with CAD undergoing PCI before the administration of contrast medium.

NGAL is a protein in the lipocalin family and is expressed by neutrophils and various epithelial cells [14]. NGAL is well known to exert a bacteriostatic effect by depleting siderophores, and on the other hand, increased serum NGAL levels have been reported in the setting of systemic disease in the absence of overt bacterial infection. Expression of NGAL increases 25- to 100-fold in humans in response to renal tubular injury and appears very rapidly in urine and serum [15]. Although the clinical usefulness of NGAL is well known in kidney injury, elevated NGAL has also been recently reported in heart failure, coronary artery disease, and cerebrovascular disease. NGAL is upregulated under conditions of failing myocardium, atherosclerotic plaques, and systemic inflammation [16]. Some investigators have suggested that serum NGAL may be of prognostic value in patients with myocardial infarction [17,18]. Here, our study has shown that baseline serum NGAL also has prognostic value in patients with CAD. Moreover, a serum NGAL level in the highest tertile was a risk factor for the occurrence of CI-AKI.

Contrast-induced AKI is an important complication of any procedure using intravascular contrast. In a retrospective analysis of the Mayo Clinic PCI registry, Rihal et al. reported the incidence among patients undergoing PCI to be 3.3%, and the 5-year estimated mortality rate in survivors with AKI was 44.6% [19]. In another large-scale PCI registry, Tsai et al. showed that 7.1% of patients experienced AKI and 0.3% required the initiation of dialysis. The risks of in-hospital myocardial infarction, bleeding, and death were greater for patients who had AKI after undergoing PCI than for those who did not have AKI [1]. To date, there have been no clinical trials demonstrating the prevention of CI-AKI. N-acetylcysteine (NAC) is a pharmacological drug that has been most widely studied in randomized controlled trials. Recent work by Weisbord et al. showed no benefit of oral NAC over placebo for the prevention of CI-AKI in the PRESERVE trial [20]. In previous studies, older age, left ventricular systolic dysfunction, chronic kidney disease (CKD), diabetes, acute coronary syndrome, and cardiogenic shock were associated with AKI after PCI [1,21]. In patients with high-risk factors, as mentioned above, adequate intravascular volume expansion with isotonic saline before and after contrast media exposure along with the avoidance of nephrotoxic drugs is the only recommended prophylactic strategy to date [22,23]. Our study shows that old age, female sex, diabetes, LV systolic dysfunction, CKD, and baseline NGAL levels are independent risk factors for CI-AKI. Even after adjustment for well-established risk factors, baseline serum NGAL was found to be a strong risk factor for CI-AKI. In addition to traditional risk factors, baseline serum NGAL could be considered a predictor of the occurrence of CI-AKI.

Serum creatinine and urine output are the most frequently monitored parameters of kidney injury in practice. However, they have several limitations, such as a slow rate of change, low sensitivity and specificity, and appearing relatively normal in early diabetic nephropathy [24]. A biomarker that could be validated to predict CI-AKI would be very useful for guiding treatment. For these reasons, there have been many efforts to find a biomarker that can detect CI-AKI occurrence earlier. Serum or urinary NGAL, cystatin C, beta-2 microglobulin, kidney injury molecule-1, and calprotectin have been widely investigated [25,26]. Of these biomarkers, NGAL is known to reflect renal tubular injury [6]. Creatinine cannot be used for the early detection of CI-AKI since it increases 3 to 5 days after contrast use; it can monitor only the occurrence of AKI. On the other hand, NGAL is known to be increased within 1 day after contrast use. Studies have shown that NGAL is helpful for predicting CI-AKI in patients undergoing PCI [7,8,9,27]. Liao et al. showed an increase in serum NGAL after PCI associated with contrast-induced nephropathy. Using small registry data, Nusca et al. found that changes in serum NGAL at baseline and post-PCI hastened the diagnosis and treatment of CI-AKI. However, there are no studies showing an association of baseline serum NGAL with CI-AKI and clinical outcomes. In the present study, we showed that the baseline serum NGAL level, not the change in NGAL, could be used to predict the occurrence of CI-AKI. Even if the baseline creatinine, eGFR, and contrast volume values were the same, it was confirmed that the higher the baseline NGAL was, the more likely CI-AKI was to occur. This allows us to predict CI-AKI before PCI and prepare preemptive treatment in advance. Medical interventions to prevent CI-AKI may be necessary for patients with elevated baseline serum NGAL.

Notably, 30–40% of patients with coronary artery disease undergoing PCI were reported to have concomitant CKD [28,29]. Cardiovascular mortality has been shown to be inversely proportional to the estimated glomerular filtration rate, with impaired renal function being an independent predictor of cardiovascular risk [30]. Myocardial revascularization guidelines recommend evaluating renal function and the risk of contrast-induced nephropathy [12,31]. The progression to heart failure or renal failure was associated with poor clinical outcomes in ischemic heart disease. In addition, NGAL could be considered a marker of inflammation and vascular injury in patients with heart failure or renal failure because NGAL is secreted and expressed by neutrophils, epithelial cells, renal tubular cells, and hepatic cells. In coronary artery disease, Zahler et al. reported that elevated NGAL levels were associated with adverse renal and cardiovascular outcomes in 267 STEMI patients [17]. Bulluck et al. showed that a higher preoperative serum NGAL was associated with an increased risk of postoperative AKI and 1-year mortality after coronary artery bypass graft surgery [32]. Our study showed the association between baseline serum NGAL levels and MACCEs. To our knowledge, this study is the first analysis of baseline serum NGAL as a prognostic biomarker in all-comer PCI populations in a real-world registry. Measurement of baseline serum NGAL may help to identify CI-AKI early, and there might be a role for this biomarker in guiding treatment to improve cardiovascular outcomes. It seems beneficial that early interventions to protect renal function lead to better clinical outcomes because renal dysfunction affects the prognosis in CAD patients. Therefore, serum NGAL could be considered a stratification tool for identifying patients at risk for CI-AKI prior to coronary intervention.

Our study has some limitations. First, the possibility of unmeasurable confounders and selection bias should be considered because this study used a retrospective, observational, and nonrandomized study design. Second, we did not measure urinary NGAL or serial serum NGAL after the index procedure. If the urinary NGAL and serial serum NGAL levels were measured together, we could have better identified their associations with clinical outcomes. The relationship between changes in NGAL and outcomes has already been discussed in previous studies. We believe that showing the role of baseline serum NGAL as a predictor of CI-AKI and prognosis in post-PCI patients has clinical implications. Third, the incidence of CI-AKI was higher than that in previous large-scale studies; nonetheless, there were no patients who required dialysis. The criteria for defining AKI vary from study to study. It is possible that our broader definition of acute kidney injury (absolute change in creatinine of ≥0.3 mg/dL or of ≥1.5 times from baseline or oliguria) than that in other studies was responsible for the differences in CI-AKI incidence.

## 5. Conclusions

The measurement of serum NGAL before PCI is helpful in predicting the development of contrast-induced acute kidney injury. High serum NGAL is independently associated with an increased risk for long-term clinical outcomes in patients with CAD treated by PCI. Baseline serum NGAL could be used as a stratifying biomarker to identify patients at risk for CI-AKI prior to PCI and long-term prognosis.

## Figures and Tables

**Figure 1 jcm-11-05971-f001:**
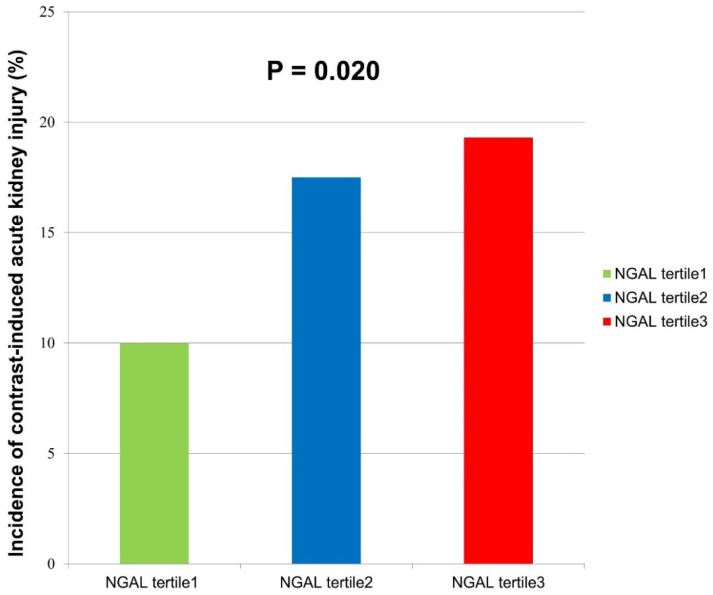
The incidence of contrast-induced acute kidney injury according to tertiles of baseline serum NGAL.

**Figure 2 jcm-11-05971-f002:**
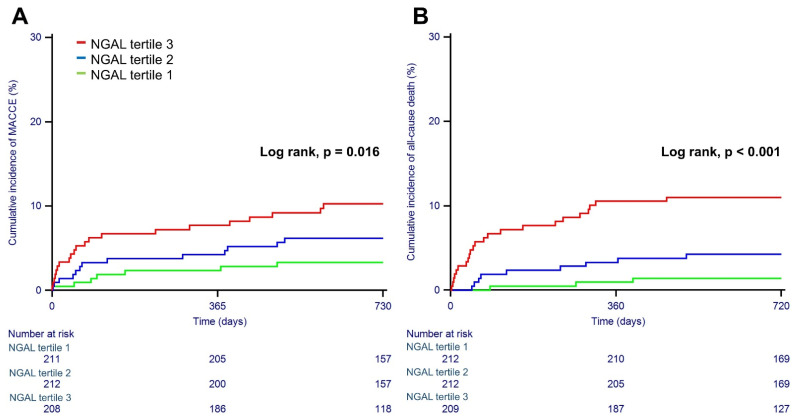
Kaplan-Meier curves for major adverse cardiac and cerebrovascular events (**A**) and all-cause death (**B**) according to tertiles of baseline serum NGAL.

**Table 1 jcm-11-05971-t001:** Baseline clinical and angiographic characteristics.

Variables	NGAL Tertile 1 (*n* = 212)	NGAL Tertile 2 (*n* = 212)	NGAL Tertile 3 (*n* = 209)	*p* Value
Age (years)	63.8 ± 10.8	65.1 ± 11.4	67.8 ± 12.7	0.002
Male	131 (61.8%)	136 (64.2%)	151 (72.2%)	0.060
Body mass index (kg/m^2^)	24.9 ± 3.1	24.7 ± 3.9	24.7 ± 3.5	0.834
Hypertension	139 (65.6%)	156 (73.6%)	161 (77.0%)	0.027
Diabetes mellitus	80 (37.7%)	70 (33.0%)	98 (46.9%)	0.012
Dyslipidemia	70 (33.0%)	76 (35.8%)	75 (35.9%)	0.777
Current smoking	53 (25.0%)	64 (30.2%)	64(30.6%)	0.363
Family history of coronary artery disease	20 (9.4%)	19 (9.0%)	15 (7.2%)	0.683
Prior stroke	17 (8.0%)	19 (9.0%)	31 (14.8%)	0.049
Prior myocardial infarction	19 (9.0%)	19 (9.0%)	16 (7.7%)	0.858
Prior percutaneous coronary intervention	29 (13.7%)	29 (13.7%)	27 (12.9%)	0.966
Prior statin use	59 (27.8%)	67 (31.6%)	66 (31.6%)	0.624
Clinical presentation				<0.001
Stable angina pectoris	73 (34.4%)	59 (27.8%)	41 (19.6%)	
Unstable angina pectoris	61 (28.8%)	51 (24.1%)	33(15.8%)	
NSTEMI	48 (22.6%)	60 (28.3%)	89 (42.6%)	
STEMI	27 (12.7%)	38 (17.9%)	40 (19.1%)	
Silent myocardial ischemia	3 (1.4%)	4 (1.9%)	6 (2.9%)	
Ejection fraction (%)	57.1 ± 10.5	54.5 ± 12.8	52.6 ± 13.0	0.001
Total cholesterol (mg/dL)	135.2 ± 34.9	137.2 ± 31.6	131.4 ± 30.8	0.414
Triglyceride (mg/dL)	135.2 ± 80.3	144.1 ± 75.9	194.5 ± 306.3	0.032
HDL cholesterol (mg/dL)	47.0 ± 11.2	45.1 ± 10.5	41.0 ± 10.0	<0.001
LDL cholesterol (mg/dL)	72.2 ± 24.4	74.6 ± 22.7	71.4 ± 21.3	0.558
High-sensitivity C-reactive protein (mg/L)	5.2 ± 17.3	7.3 ± 20.5	17.8 ± 39.5	<0.001
Creatinine (mg/dL)	0.89 ± 0.22	1.02 ± 0.32	1.46 ± 2.40	<0.001
eGFR (mL/min/1.73 m^2^)	83.1 ± 31.5	72.3 ± 27.0	58.5 ± 26.6	<0.001
eGFR <60 mL/min/1.73 m^2^	41 (19.3%)	78 (36.8%)	106 (50.7%)	<0.001
Hemoglobin (mg/dL)	13.7 ± 2.9	13.5 ± 2.0	13.2 ± 2.4	0.167
Medications at discharge				
Aspirin	210 (99.1%)	207 (97.6%)	198 (94.7%)	0.025
Clopidogrel	150 (70.8%)	131 (61.8%)	115 (55.0%)	0.004
Potent P2Y12 inhibitor	62 (29.2%)	80 (37.7%)	94 (45.0%)	0.004
Statins	210 (99.1%)	210 (99.1%)	206 (98.6%)	0.856
Beta-blocker	134 (63.2%)	151 (71.2%)	160 (76.6%)	0.011
Renin angiotensin system inhibitor	124 (58.5%)	113 (53.3%)	107 (51.2%)	0.302
Hypotension	13 (6.1%)	26 (12.3%)	26 (12.4%)	0.052
IABP or ECMO	0 (0%)	2 (0.9%)	2 (1.0%)	0.363
Culprit coronary lesion				0.215
Left anterior descending	114 (54.5%)	101 (48.1%)	91 (45.0%)	
Left circumflex	30 (14.4%)	41 (19.5%)	49 (24.3%)	
Right	53 (25.4%)	59 (28.1%)	54 (26.7%)	
Left main	12 (5.7%)	9 (4.3%)	7 (3.5%)	
Multivessel	55 (25.9%)	67 (31.6%)	82 (39.2%)	0.014
Contrast volume (mL)	198.9 ± 103.6	221.1 ± 127.9	233.1 ± 124.1	0.020
Number of total stents	1.56 ± 0.92	1.71 ± 1.00	1.89 ± 1.16	0.006
Mean diameter of stents (mm)	3.13 ± 0.43	3.12 ± 0.44	3.07 ± 0.39	0.255
Total length of stents (mm)	39.1 ± 26.9	42.9 ± 29.7	51.1 ± 36.1	<0.001

Note: Values are number (%) or mean ± standard deviation. Abbreviation: NSTEMI, non-ST-segment elevation myocardial infarction; STEMI, ST-segment elevation myocardial infarction; HDL, high-density lipoprotein; LDL, low-density lipoprotein; eGFR, estimated glomerular filtration rate; IABP, intra-aortic balloon pump; ECMO, extracorporeal membrane oxygenation.

**Table 2 jcm-11-05971-t002:** Associations between clinical characteristics and the risk of contrast-induced acute kidney injury according to univariate and multivariate logistic regression models.

	Univariate		Multivariate	
	OR (95% CI)	*p* Value	OR (95% CI)	*p* Value
Age	1.025 (1.005–1.044)	0.012	1.041 (1.014–1.068)	0.003
Female	1.556 (1.003–2.415)	0.048	0.337 (0.151–0.751)	0.008
Body mass index	0.953 (0.879–0.994)	0.030	0.987 (0.919–1.060)	0.716
Hypertension	1.245 (0.757–2.046)	0.388		
Diabetes mellitus	1.714 (1.111–2.644)	0.015	1.787 (1.082–2.952)	0.023
Dyslipidemia	0.560 (0.342–0.918)	0.021	0.361 (0.076–1.701)	0.198
Smoking	1.134 (0.709–1.813)	0.599		
Family history of CAD	0.813 (0.356–1.857)	0.623		
Chronic kidney disease	1.730 (0.759–3.942)	0.192		
Prior stroke	1.261 (0.647–2.458)	0.496		
Prior statin use	0.622 (0.374–1.034)	0.067	1.952 (0.395–9.648)	0.412
Acute myocardial infarction	1.752 (1.131–2.714)	0.012	1.618 (0.962–2.721)	0.069
Left ventricular ejection fraction	0.963 (0.948–0.979)	<0.001	0.961 (0.943–0.980)	<0.001
eGFR	1.012 (1.005–1.018)	0.001	1.035 (1.023–1.047)	<0.001
Hemoglobin	0.900 (0.810–0.999)	0.048	0.932 (0.814–1.068)	0.310
Multivessel disease	1.258 (0.803–1.972)	0.317		
LAD lesion	1.451 (0.930–2.262)	0.101		
Hypotension	0.877 (0.418–1.839)	0.728		
Contrast volume	1.000 (0.998–1.002)	0.816		
NGAL tertile 2	1.913 (1.078–3.394)	0.027	2.700 (1.391–5.239)	0.003
NGAL tertile 3	2.167 (1.228–3.823)	0.008	3.573 (1.788–7.141)	<0.001

OR indicates odds ratio; CI, confidence interval; CAD, coronary artery disease; eGFR, estimated glomerular filtration rate; LAD, left anterior descending.

**Table 3 jcm-11-05971-t003:** Clinical outcomes according to NGAL tertiles.

	NGAL			*p* Value
	Tertile 1	Tertile 2	Tertile 3	
MACCEs	8 (3.8%)	13 (6.1%)	22 (10.5%)	0.020
All-cause death	3 (1.4%)	9 (4.2%)	23 (11.0%)	<0.001
Cardiovascular death	2 (0.9%)	7 (3.3%)	15 (7.2%)	0.003
Nonfatal myocardial infarction	1 (0.5%)	1 (0.5%)	0 (0%)	0.610
Nonfatal stroke	2 (0.9%)	2 (0.9%)	2 (1.0%)	0.999
Any revascularization	5 (2.4%)	5 (2.4%)	6 (2.9%)	0.928

MACCEs, major adverse cardiac and cerebrovascular events; NGAL, neutrophil gelatinase-associated lipocalin.

**Table 4 jcm-11-05971-t004:** Hazard ratios of baseline serum NGAL tertiles for MACCEs and all-cause death.

	Model 1		Model 2		Model 3		Model 4	
	HR (95% CI)	*p* Value	HR (95% CI)	*p* Value	HR (95% CI)	*p* Value	HR (95% CI)	*p* Value
MACCEs								
Tertile 1	1		1		1		1	
Tertile 2	1.652 (0.685–3.986)	0.264	1.600 (0.662–3.868)	0.297	1.545 (0.633–3.775)	0.340	2.151 (0.827–5.592)	0.116
Tertile 3	2.984 (1.328–6.704)	0.008	2.781 (1.211–6.386)	0.016	2.596 (1.093–6.167)	0.031	2.725 (1.052–7.058)	0.039
All-cause death								
Tertile 1	1		1		1		1	
Tertile 2	3.039 (0.823–11.227)	0.095	2.631 (0.710–9.752)	0.148	2.437 (0.650–9.142)	0.187	3.692 (0.938–14.522)	0.062
Tertile 3	8.260 (2.480–27.512)	0.001	5.879 (1.721–20.078)	0.005	5.077 (1.416–18.201)	0.013	6.172 (1.650–23.077)	0.007

Model 1 is the univariate analysis. Model 2 is adjusted for age and sex. Model 3 is adjusted for age, sex, and estimated glomerular filtration rate. Model 4 is adjusted for age, sex, estimated glomerular filtration rate, body mass index, hypertension, diabetes mellitus, smoking, family history of coronary artery disease, chronic kidney disease, dyslipidemia, prior stroke, prior myocardial infarction, acute myocardial infarction, hypotension, multivessel disease, and left ventricular ejection fraction. MACCEs, major adverse cardiac and cerebrovascular events, HR, hazard ratio, CI, confidence interval.

## Data Availability

The data presented in the current study are available on reasonable request from the corresponding author.

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
