# Peer review of "Predictive and Prognostic Value of Serum Neutrophil Gelatinase-Associated Lipocalin for Contrast-Induced Acute Kidney Injury and Long-Term Clinical Outcomes after Percutaneous Coronary Intervention"

_jcm, 2022, doi:10.3390/jcm11195971_

Round 1
Reviewer 1 Report
The authors present data NGAL as a marker for acute renal injury following percutaneous coronary intervention. This is an interesting study of consecutive patients with coronary artery disease.
This reviewer would like the authors to address the following points:
1) How does hypotension during the procedure control for the NGAL levels. Does hypotension, use of balloon pump or ECMO affect NGAL levels independent of use of contrast?
2) Many pts with CAD have CKD with normal creatinine, especially when they have diabetes. How would the authors account for pre-existing renal disease in these patients?
Author Response
Reviewer 1.
The authors present data NGAL as a marker for acute renal injury following percutaneous coronary intervention. This is an interesting study of consecutive patients with coronary artery disease.
This reviewer would like the authors to address the following points:
1) How does hypotension during the procedure control for the NGAL levels. Does hypotension, use of balloon pump or ECMO affect NGAL levels independent of use of contrast?
Response 1> Thank you for your constructive comments. There might be a correlation between hypotension with baseline NGAL levels because the group of 2nd and 3rd NGAL tertiles have more hypotension. We thought that the number of patients treated IABP or ECMO was too small to figure out a statistical relationship in our study.
2) Many pts with CAD have CKD with normal creatinine, especially when they have diabetes. How would the authors account for pre-existing renal disease in these patients?
Response 2> Thank you for your good comments. As you pointed out, creatinine is often normal in patients with early diabetic nephropathy. This is a disadvantage of serum creatinine. Much research are being done to find other novel biomarkers. We believe that NGAL may reflect renal disease that creatinine did not reflect in this patient group. Therefore, the results of this study are considered to be meaningful.
Modifications
(Discussion)
However, they have several limitations, such as a slow rate of change and low sensitivity and specificity. [24]
However, they have several limitations, such as a slow rate of change, low sensitivity and specificity, and relatively normal in early diabetic nephropathy. [24]

Reviewer 2 Report
Byeon and colleagues explored the potential of circulating NGAL as a biomarker of Contrast-Induced Acute Kidney Injury (CI AKI). They stratified patients according to the basal level of NGAL they measured using ELISA method in the serum of the patients before percutaneous coronary intervention (PCI) into tertiles and observed that the rate of major adverse cardiac and cerebrovascular events (MACCES) was higher in patients with high baseline NGAL concentration.
The present study is well written and the data support the authors' conclusions. The method is very precise and clear and the number of samples (>600) is impressive.
Minors comments:
-PCI is abbreviated in the abstract but is not used anymore in the abstract. In contrast, CAD abbreviation is not developed in the abstract. Could the authors correct this discrepancy?
-The main method used here is ELISA. Nevertheless, the authors did not specify the catalogue reference of the product. On R&D Systems website, 5 references correspond to human NGAL. The authors must precise which product they used in order to facilitate the reproduction of their findings.
-Univariate analysis is not detailed in the Material section. How did the authors choose the variables the tested in the multivariate analysis? Usually, once considers that all variables with p<0.2 in univariate test are subjected to multivariate analysis. Could the authors be more specific on this particular point? (For instance, prior stroke proportion is significantly increase in the 3rd tertile but it was not included in multivariate analysis)
-A paragraph in the discussion section dedicated to the biological cellular origine and a suggestion of the biological mechanism induced by NGAL in this particular context could be very appreciated.
Author Response
Reviewer 2.
Byeon and colleagues explored the potential of circulating NGAL as a biomarker of Contrast-Induced Acute Kidney Injury (CI AKI). They stratified patients according to the basal level of NGAL they measured using ELISA method in the serum of the patients before percutaneous coronary intervention (PCI) into tertiles and observed that the rate of major adverse cardiac and cerebrovascular events (MACCES) was higher in patients with high baseline NGAL concentration.
The present study is well written and the data support the authors' conclusions. The method is very precise and clear and the number of samples (>600) is impressive.
Minors comments:
-PCI is abbreviated in the abstract but is not used anymore in the abstract. In contrast, CAD abbreviation is not developed in the abstract. Could the authors correct this discrepancy?
Response 1> Thank you for your good comments. We deleted an abbreviated PCI and used the full medical terminology of CAD. (Coronary artery disease)
Modifications
(Abstract)
Serum NGAL was measured before percutaneous coronary intervention (PCI) in 633 patients with CAD.
Serum NGAL was measured before percutaneous coronary intervention in 633 patients with coronary artery disease.
-The main method used here is ELISA. Nevertheless, the authors did not specify the catalogue reference of the product. On R&D Systems website, 5 references correspond to human NGAL. The authors must precise which product they used in order to facilitate the reproduction of their findings.
Response 2> Thank you for this constructive comment. We corrected catalogue reference of NGAL.
Modifications
(Materials and Methods)
Serum NGAL levels were measured by an enzyme-linked immunosorbent assay (ELISA) kit from R&D Systems (Minneapolis, Minnesota).
Serum NGAL levels were measured by a Human Lipocalin-2/NGAL Quantikine ELISA kit (Catalog #DLCN20) from R&D Systems (Minneapolis, Minnesota).
-Univariate analysis is not detailed in the Material section. How did the authors choose the variables the tested in the multivariate analysis? Usually, once considers that all variables with p<0.2 in univariate test are subjected to multivariate analysis. Could the authors be more specific on this particular point? (For instance, prior stroke proportion is significantly increase in the 3rd tertile but it was not included in multivariate analysis)
Response 3> We deeply thank the reviewer for this important comment. In Table 2, traditional cardiovascular risk factors and CI-AKI risk factors were used for univariate analysis, and only variables with p < 0.1 in univariate analysis were analyzed by multivariate analysis.
In Table 4, multivariate analysis was performed using traditional risk factors related to cardiovascular outcomes. Based on your valuable comments, we performed multivariate analysis again by adding prior stroke, hypotension, and multivessel disease. New Table 4 presents the revised statistical values.
Modifications
(Abstract)
Using the first tertile as a reference, the adjusted hazard ratios for MACCEs in patients in the second and third tertiles of NGAL were 1.977 (confidence interval (CI) 0.79 to 4.92, p=0.143) and 2.586 (CI 1.06 to 6.30, p=0.037), respectively.
Using the first tertile as a reference, the adjusted hazard ratios for MACCEs in patients in the second and third tertiles of NGAL were 2.151 (confidence interval (CI) 0.82 to 5.59, p=0.116) and 2.725 (CI 1.05 to 7.05, p=0.039), respectively.
(Methods)
Multivariable analysis was performed to assess the prognostic value of serum NGAL and MACCEs after adjusting for age, sex, estimated glomerular filtration rate, body mass index, hypertension, diabetes mellitus, smoking, family history of coronary artery disease, chronic kidney disease, dyslipidemia, prior myocardial infarction, acute myo-cardial infarction, and left ventricular ejection fraction.
Traditional cardiovascular risk factors and CI-AKI risk factors were used for univariate analysis and only variables with p < 0.1 in univariate analysis were analyzed by multivariate analysis for association with the risk of contrast-induced acute kidney injury. Multivariable analysis was performed to assess the prognostic value of serum NGAL and MACCEs after adjusting for age, sex, estimated glomerular filtration rate, body mass index, hypertension, diabetes mellitus, smoking, family history of coronary artery disease, chronic kidney disease, dyslipidemia, prior stroke, prior myocardial in-farction, acute myocardial infarction, hypotension, multivessel disease, and left ven-tricular ejection fraction.
(Results)
Using the first tertile as a reference, the adjusted HRs for those in the second and third ter-tiles of NGAL were 1.977 (CI 0.794-4.924, p=0.143) and 2.586 (CI 1.061-6.302, p=0.037) for MACCEs and 3.097 (CI 0.777-12.351, p=0.109) and 4.946 (CI 1.288-19.002, p=0.020) for all-cause death (Table 4).
Please see the attachment
Using the first tertile as a reference, the adjusted HRs for those in the second and third ter-tiles of NGAL were 2.151 (CI 0.827-5.592, p=0.116) and 2.725 (CI 1.052-7.058, p=0.039) for MACCEs and 3.692 (CI 0.938-14.522, p=0.062) and 6.172 (CI 1.650-23.077, p=0.007) for all-cause death (Table 4).
-A paragraph in the discussion section dedicated to the biological cellular origine and a suggestion of the biological mechanism induced by NGAL in this particular context could be very appreciated.
Response 4> Thank you for this constructive comment. We added biological cellular origin and biological mechanism induced by NGAL in discussion section.
Modifications
(Discussion)
NGAL is well known to exert a bacteriostatic effect by depleting siderophores, and on the other hand, increased serum NGAL levels have been reported in the setting of systemic disease in the absence of overt bacterial infection. Expression of NGAL increases 25 to 100-fold in humans in response to renal tubular injury and appears very rapidly in urine and serum.[15]

Reviewer 3 Report
Dear Authors,
Very well conducted observational study. Explained the results and limitations of the study well.
Please address these in the manuscript if available
It would be useful for the readers if you can add the thoughts about what might be the possible medical interventions that could be done after knowing that a particular patient has elevated baseline NGAL and is at increased risk for CI-AKI?
Do you know any studies that checked if the NGAL levels decrease with IV hydration prior to Contrast use and will that be useful in decreasing the incidence of CI-AKI in these patient with basleine elevated NGAL?
Is there any role for NGAL as a causative factor for CI-AKI?
Thank you.
Author Response
Reviewer 3.
Dear Authors, 
Very well conducted observational study. Explained the results and limitations of the study well. 
Please address these in the manuscript if available
It would be useful for the readers if you can add the thoughts about what might be the possible medical interventions that could be done after knowing that a particular patient has elevated baseline NGAL and is at increased risk for CI-AKI?
Response 1> Thank you for this constructive comment. There is no proven treatment except hydration so far. We added that medical intervention may be required in patients with high baseline NGAL.
Modifications
(Discussion)
It is thought that medical interventions to prevent CI-AKI may be necessary for patients with elevated baseline serum NGAL.
Do you know any studies that checked if the NGAL levels decrease with IV hydration prior to Contrast use and will that be useful in decreasing the incidence of CI-AKI in these patient with basleine elevated NGAL?
Response 2> Thank you for your good comments. However, to our knowledge, we could not find any study in which IV hydration before contrast use reduced serum baseline NGAL levels. We thought further studies on the clinical effect(CV outcomes or CI-AKI occurrence) of IV hydration before contrast use in patients with high baseline NGAL may be necessary.
Is there any role for NGAL as a causative factor for CI-AKI?
Response 3> We thank you for your good comments. We are very sorry for not finding a role for NGAL as a causative factor for CI-AKI from the literature review.
Please see the attachment.

Reviewer 4 Report
CI-AKI is one of the important complications that could affect prognosis and quality of life for patients needing percutaneous diagnostic and therapeutical intervantions in CAD and not only...
I appreciate the study design, the categorization in three groups according to the tertile of baseline serum NGAL, as well as the univariate and multivariate analysis.
Please, pay attention to page 5, where Table S1, that I have markedwith yellow in the text, was supposed to contain informations about the 98 patients (15.5%) that developed CI-AKI.
Also, you did not mention, or I could not find the information about % of severe AKI pacients needing kidney replacement therapy, CVVHDF, for example.
Also, if you discuss the CLINICAL outcomes in patients with AKI, I believe you should add data about persistent/aggravated kidney disfunction, renal death and evolution to chronic dialysis.
Suggested references:
1. Andrew Kei-Yan Ng, Pauline Yeung Ng, April Ip, Lap-tin Lam, Ian Wood-Hay Ling, Alan Shing-Lung Wong, Desmond Yat-Hin Yap, Chung-Wah Siu, Impact of contrast-induced acute kidney injury on long-term major adverse cardiovascular events and kidney function after percutaneous coronary intervention: insights from a territory-wide cohort study in Hong Kong, Clinical Kidney Journal, Volume 15, Issue 2, February 2022, Pages 338–346, https://doi.org/10.1093/ckj/sfab212
2. Bucaloiu ID, Kirchner HL, Norfolk ER et al. Increased risk of death and de novo chronic kidney disease following reversible acute kidney injury. Kidney Int 2012; 81: 477–485
Congratulations for your work !
Author Response
Reviewer 4.
CI-AKI is one of the important complications that could affect prognosis and quality of life for patients needing percutaneous diagnostic and therapeutical intervantions in CAD and not only...
I appreciate the study design, the categorization in three groups according to the  tertile of baseline serum NGAL, as well as the univariate and multivariate analysis. 
Please, pay attention to page 5, where Table S1, that I have marked with yellow in the text, was supposed to contain informations about the 98 patients (15.5%) that developed CI-AKI.
Response 1> Thank you for this constructive comment. However, we could not find your yellow marking in the text in page 5. Do you mean to add clinical information of 98 patients who developed CK-AKI?
Also, you did not mention, or I could not find the information about % of severe AKI patients needing kidney replacement therapy, CVVHDF, for example. 
Also, if you discuss the CLINICAL outcomes in patients with AKI, I believe you should add data about persistent/aggravated kidney disfunction, renal death and evolution to chronic dialysis.
Response 2> We deeply thank the reviewer for this important comment. In our study, there were no reports of patients requiring renal replacement therapy during hospitalization or full follow-up. We added this to the results section for readers to know. We thought this point is one of the limitations of our study. So, it has already been mentioned in the limitation sections.
Modifications
(Results)
Patients who required renal replacement therapy were not reported during in-hospital periods and overall follow-up.
Suggested references:
- Andrew Kei-Yan Ng, Pauline Yeung Ng, April Ip, Lap-tin Lam, Ian Wood-Hay Ling, Alan Shing-Lung Wong, Desmond Yat-Hin Yap, Chung-Wah Siu, Impact of contrast-induced acute kidney injury on long-term major adverse cardiovascular events and kidney function after percutaneous coronary intervention: insights from a territory-wide cohort study in Hong Kong, Clinical Kidney Journal, Volume 15, Issue 2, February 2022, Pages 338–346, https://doi.org/10.1093/ckj/sfab212
- Bucaloiu ID, Kirchner HL, Norfolk ER et al. Increased risk of death and de novo chronic kidney disease following reversible acute kidney injury. Kidney Int 2012; 81: 477–485
Response 3> Thank you for this constructive suggestion. We added your suggested references.
Modifications
(Manuscript)
Contrast-induced acute kidney injury (CI-AKI) is a major complication of coronary artery disease (CAD) treated by percutaneous coronary intervention (PCI) [1] and is associated with increased mortality. [2]
Contrast-induced acute kidney injury (CI-AKI) is a major complication of coronary artery disease (CAD) treated by percutaneous coronary intervention (PCI) [1] and is associated with increased mortality and cardiovascular outcomes. [2-4]
(References)
- Ng, A. K.; Ng, P. Y.; Ip, A.; Lam, L. T.; Ling, I. W.; Wong, A. S.; Yap, D. Y.; Siu, C. W., Impact of contrast-induced acute kidney injury on long-term major adverse cardiovascular events and kidney function after percutaneous coronary intervention: insights from a territory-wide cohort study in Hong Kong. Clin Kidney J 2022, 15 (2), 338-346.
- Bucaloiu, I. D.; Kirchner, H. L.; Norfolk, E. R.; Hartle, J. E., 2nd; Perkins, R. M., Increased risk of death and de novo chronic kidney disease following reversible acute kidney injury. Kidney Int 2012, 81 (5), 477-85.
